# The Analysis of Dental Treatment under General Anaesthesia in Medically Compromised and Healthy Children

**DOI:** 10.3390/ijerph16142528

**Published:** 2019-07-15

**Authors:** Romana Koberova Ivancakova, Jakub Suchanek, Flora Kovacsova, Eva Cermakova, Vlasta Merglova

**Affiliations:** 1Department of Dentistry, Faculty of Medicine Charles University and University Hospital, 500 03 Hradec Kralove, Czech Republic; 2Department of Medical Biophysics, Faculty of Medicine Charles University, 500 03 Hradec Kralove, Czech Republic; 3Department of Dentistry, Faculty of Medicine Charles University and University Hospital, 323 00 Pilsen, Czech Republic

**Keywords:** dental treatment, general anaesthesia, children, medically compromised

## Abstract

Dental care under general anaesthesia (GA) is an option when normal treatment cannot be accomplished due to un-cooperation and systemic or cognitive/intellectual disabilities. The purpose of this retrospective cohort study was to analyse the dental treatment under GA in medically compromised and healthy children. The data were collected from the medical records of children who received their dental treatment under GA. The data regarding patient age, sex, general health, and type of treatment were analysed. This clinical trial included 229 study subjects (138 males, 91 females) with an average age of 8.34 (SD 3.78). Counts and relative counts were used for description of qualitative data. The association between the variables was analysed using contingency tables. The significance of the findings was tested by the chi-square test. Most of the children were older pre-school 63 (27.51%) and young school children 102 (44.54%). Medical disability (systemic or intellectual) was diagnosed in 142 children (62.01%); the remaining 87 (37.99%) were healthy children. Dental treatment of primary teeth was more commonly performed in healthy children (65.52%) compared to medically compromised children (58.45%) (*p* = 0.287). The total number of medically compromised children and the total number of healthy children were both considered to be 100% for the purpose of the following calculations. In terms of permanent dentition, medically compromised children required more extractions and fillings (38.03%, 57.04%) compared to healthy children (14.94%, 17.24%, respectively). The results of this study revealed that dental treatment under GA was more commonly performed in medically compromised children in permanent teeth only in comparison to healthy children. Based on these findings, both health professionals and state authorities should focus more on preventive care in medically compromised children in order to improve their oral health.

## 1. Introduction

Dental care under general anaesthesia (GA) is an option when normal treatment cannot be accomplished due to young age of the child or due to systemic chronic disease and psychological disabilities. Factors affecting the decision on dental treatment under GA or traditional dental care include the quality and quantity of treatment need, the child′s age, and the child’s cooperation [1]. Although the use of special behaviour management techniques and sedation are encouraged for challenging patients, some interventions may not be addressed by behaviour guidance alone due to extensive dental treatment, complexity, and medical conditions. In these cases, GA is a reliable tool for safe and successful dental treatment, thus allowing for the provision of high-quality comprehensive dental care. This is especially true for very young and medically compromised children [2,3,4]. Medically compromised children are well known to frequently present with unmet dental treatment needs that require the use of GA [5]. Medically compromised children are defined as children with any physical, developmental, mental, sensory, behavioural, cognitive, or emotional impairment [6]. With respect to oral manifestation, most of these patients have a higher rate of dental caries and bacterial plaque in both primary and permanent teeth due to a variety of factors, including difficulty controlling oral hygiene, a soft and high sugar containing diet, difficulties in chewing and swallowing, and medication [7]. Medically compromised individuals may be at an increased risk of oral diseases through their lifetime [8]. The data for this study were collected in Czech Republic, where public health service does not provide any systematic specialized dental care for medically compromised individuals. The null hypothesis of this study was that dental treatment under GA is independent of general health status. 

The purpose of this study was to find out if the need of dental treatment under GA is related to general health status. 

## 2. Materials and Methods

The data for this study were collected from the medical records of children who had been treated under GA at the Department of Dentistry and Department of Paediatric Surgery, Faculty of Medicine, Charles University and University Hospital in Hradec Kralove (Czech Republic) within the last four years. Inclusion criteria were children (both medically compromised and healthy children) with extensive oral health problems who needed dental treatment under GA. The study was included in the research project of Charles University in Prague PROGRES Q29 and approved by the Ethical Committee, Faculty of Medicine, Charles University in Hradec Kralove no. 201803 S14P. The child was considered medically compromised when he/she had a diagnosis of any medical disorder (International Classification of Diseases 10). The standard written informed consent for treatment under GA was obtained from parents or caregivers prior to treatment. The study included 229 subjects (138 males, 91 females; a male to female ratio of 1.52:1) with an average age of 8.34 and a median age of 8 years (interquartile range 5–11). The patients were either treated on an outpatient basis or were receiving stationary, perioperative, and postoperative care and monitoring (heart rate, blood pressure, oxygen saturation, breathing frequency), including pain control in a paediatric hospital setting. Ambulatory surgery was indicated for ASA (American Association of Anaesthesiologists) I and ASA II patients, while ASA III patients were more likely to be hospitalized due to complexity of their underlying disease [9]. Paediatric dentists had treated all children using the same operating units. 

The data regarding patient age, sex, general health, and dental treatment (restorations or extractions) were collected and analysed.

All the data were collected and sorted using the spreadsheet programme Microsoft Office, Excel 2007, Redmond, Washington, U.S. and were subsequently exported into NCSS 11 Statistical Software (Kaysville, Utah, U.S) for descriptive analyses and further testing. The data were not normalized.

Median and interquartile range were used for description of age. Counts and relative counts were used for description of qualitative data. The association between the variables was analysed using contingency tables. The significance of the findings was tested by the chi-square test. 

## 3. Results

The study included 229 subjects (138 males, 91 females; a male to female ratio 1.52:1) with an average age of 8.34 years and a median age of 8 years (interquartile range 5–11). Based on age distribution, 19 children (8.30%) were younger pre-school children (toddlers), 63 children (27.51%) were older pre-school children, 102 (44.54%) were school children, and the remaining 45 children (19.65%) were adolescents.

Medical disability (systemic or intellectual) was diagnosed in 142 (62.01%) subjects. The most frequent medical impairments are summarized in Table 1. 

The most common dental diagnosis was dental caries and its complications, which accounted for 186 (81.22%) of cases. Other medical conditions when treatment under GA was indicated were supernumerary and impacted teeth (29, 11.79%), benign mucosal lesions (10, 4.37%), and traumatic injuries (6, 2.62%). Children with diagnoses other than dental caries were mostly healthy (97.82%).

### 3.1. Treatment of Primary Dentition

Caries-related fillings or extraction were the most common required care. Of the 229 children in the study, 140 (61.14%) children required treatment of primary teeth (restoration, extraction, or both). Dental treatment of primary teeth, Table 2, was higher but not significant in healthy children (65.52%) compared to medically compromised children (58.45%). We did not find an impact of health conditions on dental treatment in primary teeth (*p* = 0.287). Extractions of primary teeth were conducted in 132 children and occurred more frequently in healthy children (62.07%) compared to medically compromised children (54.93%) (*p* = 0.289). Restorations of primary teeth were the treatment option in 87 children, 49.43% of healthy children and 30.99% of medically compromised children (*p* = 0.00526). 

### 3.2. Treatment of Permanent Dentition

Dental treatment of permanent teeth (restoration, extraction or both), Table 3, was performed in 117 (51.09%) of children, 66.20% were medically compromised, 26.44% were healthy. We have found the impact of health conditions on the need of treatment in permanent dentition (*p* < 0.001). Extractions of permanent teeth have been performed in 67 children, 38.03% were medically compromised and 14.94% were healthy (*p* = 0.00019). Permanent teeth restorations were conducted in 96 (41.92%) of children, the most of them was medically compromised (57.04%). Only 17.24% healthy children required restorations in permanent teeth (*p* < 0.001). The number of all medically compromised children was calculated as 100% and the number of all healthy children was calculated as 100%. The indication of dental treatment under GA in permanent dentition is probably more dependent on health conditions regarding both fillings and extractions.

Some children required both restorative treatment and extractions in one sitting, thus explaining why the sum of restorations and extractions cannot be 100%. The treatment of both primary and permanent teeth (meaning mixed dentition) in one sitting was performed in 58 children (25.33%) from our study group.

## 4. Discussion

This study presents useful information about dental treatment of medically compromised and healthy individuals of the Czech paediatric population under GA over a time period of up to four years. The use of GA is valuable for young patients who require extensive dental treatment and who do not cooperate with the use of behaviour management, local anaesthesia, or sedation [10,11,12]. The second and significant group of candidates for GA are patients with physical/psychological disability and medically compromised children, due to their underlying medical conditions and the necessity of postoperative monitoring [13,14,15]. 

In total, f 229 children (both medically compromised and healthy children) received dental treatment under GA at the University Hospital, Hradec Kralove, within the last four years. In this study, the children were mostly aged from 4 to 12 years (72.05%). There was an increasing trend in the occurrence of dental treatment under GA in the study period. This might be due to the reluctance of some dentists to treat non-cooperative children, both healthy and those with medical disability, conventionally in their dental office, preferring to instead refer the children for GA [15]. Another reason might be that parents have become more aware and more demanding of this treatment option.

The dental treatment of children with disabilities and chronic illnesses are reported to be high, especially with regard to caries-related needs [16,17]. In this study, 62.01% of the children were diagnosed with systemic chronic disease or intellectual disability. Richards et al. [18] proposed dental caries to be the most common cause of dental treatment under GA, and this is in concordance with the current findings. Caries-related fillings or extractions were the most common care in both primary and permanent dentition for our study population. The treatment of dental caries and its complications in primary dentition was more frequently performed in healthy children. This might be explained by the fact that indication for GA in small healthy children is not only due to health problems but more likely due to a lack of cooperation and/or the extent of treatment. In terms of permanent dentition, medically compromised children required more extractions and restorations in comparison to healthy children. We found an impact of health conditions on dental treatment in permanent dentition. The present study findings correspond with the similar study of Rajavaara et al. [15] but are partly contradictory to the findings of Loyola-Rodrigues et al. [19], who reported more treatment need among healthy children compared to medically compromised ones.

Some of the procedures performed under GA were extensive, and comprehensive treatment can be mentally traumatic for the child and parents, particularly when a treatment involves multiple extractions. The situation after GA may also be fruitful in terms of motivating parents to promote better home care of their child′s oral health [20]. Individually designed preventive approaches to control caries after the dental treatment under GA must be emphasised in order to avoid repeated need of GA. It must be borne in mind that caries history predicts future caries incidence independently of whether the child is healthy or medically compromised. This is particularly important in terms of children with multiple health issues [15,21]. The present study, similar to the study by Rajavaara [15], did not have detailed information concerning the dental treatment history of the children. It would be interesting to know how many visits and attempts preceded the treatment under GA, reasons for referral, and reasons for the use of GA, such as the ability to cooperate. 

## 5. Conclusions

The results of this study revealed that dental treatment under GA was more commonly performed in medically compromised children compared to healthy children but only in permanent teeth. The same was not confirmed in primary teeth. Further research is necessary to study more detailed dependence of health conditions on dental treatment. Based on these findings, both health professionals and state authorities should focus more on preventive care in medically compromised children in order to improve the oral health of this group of Czech paediatric population.

## Figures and Tables

**Table 1 ijerph-16-02528-t001:** The distribution of medical disabilities.

Health Conditions	*N*	%
Mental/behaviour disorders	104	73.24
Congenital/chromosomal malformations	10	7.04
Congenital heart defects	9	6.35
Epilepsy	3	2.11
Sensory failures	3	2.11
Endocrine disorders	3	2.11
Others	10	7.04
Total	142	100.00

**Table 2 ijerph-16-02528-t002:** The percentage primary dentition treatment need.

	Healthy Children (%)	Medically Compromised Children (%)	*p*-value
Treatment	65.52	58.45	0.287
Restorations	49.43	30.99	0.00526 *
Extractions	62.07	54.93	0.289

* Statistically significant (*p* < 0.05). The chi-square statistics is 21,814. The *p*-value is <0.0001. The result is significant at *p* < 0.05.

**Table 3 ijerph-16-02528-t003:** The percentage permanent dentition treatment need.

	Healthy Children (%)	Medically Compromised Children (%)	*p*-value
Treatment	26.44	66.20	<0.001 *
Restorations	17.24	57.04	<0.001 *
Extractions	14.94	38.03	0.00019 *

* Statistical significance (*p* < 0.05). The chi-square statistics is 21.814. The *p*-value is <0.0001. The result is significant at *p* < 0.05.

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
