# Peer review of "The Analysis of Dental Treatment under General Anaesthesia in Medically Compromised and Healthy Children"

_ijerph, 2019, doi:10.3390/ijerph16142528_

Round 1

Reviewer 1 Report

In the R1 version of the manuscript entitled: “The analysis of dental treatment under general anaesthesia in medically compromised and healthy children” the authors followed all the issues suggested by the reviewer. Though the changes based on the reviewer comments, almost of the criticisms were carefully analysed and solved.

I have carefully evaluated all parts of the manuscript. I believe that the article, in this version, is now adequate for publication in this journal.

Author Response

Dear reviewer. Thank you very much for all comments and suggestions.

Reviewer 2 Report

It is not clear what changes has been made in the final submission?

The table 1 and 2 can be summarised in the result section

There are many typos and they need correction and revision

The discussion is largely repeating the result section and is very deficient , please update and compare with new studies 

Author Response

Dear reviewer. Thank you very much for all comments and suggestions. I did try to comply with them.

You are right. I      did not highlight the changes in the revised manuscript. But the suggested      changes have been made according you suggestions.

I agree the result      section may seem too long. But I feel as important to add both tables in      this section just for quick orientation.

I agree the discussion      repeating the results. But I feel it as a common issue. I have read many      scientific articles designed in such way. I´ve tried to update the      discussion section and make it more acceptable.

The English was      revised by the native English speaking colleague at our faculty of      Medicine.

In introduction section I tried to explain better why we decided for this study. This study comes up from country with different system of health care. We do not have any public health service where patients with special needs are regularly checked and medically compromised children may not achieve dental treatment in time.

Reviewer 3 Report

Though the author has tried to revise the manuscript, some critical points fail to be answered or satisfied.

1.     First and foremost:

Have this study obtained the approval of IRB? If so, please add the information.

Author’s answer: No.

All studies involved with human being should maintain IRB approval. Because this study didn’t have IRB, I am afraid that this study didn’t meet the requirement of research ethics.

2.     The study aim: The purpose of this study was to analyze the dental treatment under GA in medically compromised and healthy children.

Though the author has revised the aim, it’s still not clear. The measuring outcome cannot be understood even in the aim.

3.     The study design: The age range in the study included infants, preschoolers, children and teenagers whose dental condition varied significantly. ….. Also, the following reporting guideline is also inappropriate (first sentence in the materials and methods).

Author’s answer: Was revised and better formulated.

However, the revision failed to clarify the study design. I will suggest the author to figure out the types of observational study and their critical points including but not limited to: What is case-control study and what is it measuring, or what is cross-sectional study and how the data is presented.

Besides, the author failed to follow any reporting guidelines.

4.     The measuring outcome: What are treatment need and general health? How to define them?

Author’s answer: It was reformulated.

After going through the manuscript, the definition of ‘treatment need’ still can not be found, and it’s still unclear what is comparing in table 3 and 4 about the ‘treatment need’ column. As I have mentioned in the last review, because all of the recruited children had received treatment under GA, they all had been treated.

5.     Result:

Because the author described the treatment of primary teeth and permanent teeth, it is essential to categorize the data according to their age.

Author’s answer: It is not essential. It is not true that specific age always corresponds to specific number of teeth.

The reviewer and the author cannot meet an agreement. To the reviewer, 0-18y including primary dentition, mixed dentition, and permanent dentition. Children’s behavior and their acceptance are also different in different stage.

As the author failed to group the patients, the data presented is not correct. First of all, session 3.1 and 3.2 are primary dentition and permanent dentition. May I ask the author, how about the mixed dentition? Besides, in the second sentence of session 3.1, as this session is about primary dentition (not mixed dentition, not permanent dentition), I am afraid that 61.14% from 140/229 (229 is 0-18 year old) is meaningless. For the 18, they have zero chance to be treated with the primary dentition, isn’t it? That’s why we suggest the author to group the patient according to their age.

6.     How to control the bias?

Author’s answer: The bias was not controlled. All children treated under GA were included.

We understand that all studies have the bias and no study is perfect, but we still need to minimize it. It seems the author didn’t attempt at all to reduce it or even discuss about that.

7.     Conclusion:

‘The results support our hypothesis that dental treatment of permanent teeth under GA is more commonly performed in medically compromised children in comparison to healthy children.’

Please check the null hypothesis in the introduction. The null hypothesis is with a two-side test, but this statement of the hypothesis is one-sided. The interpretation of the hypothesis is not correct. The conclusion should match the aim instead of hypothesis. 

8.     There are some minor points:

8.1 Last paragraph of the result: Some children required both restorative treatment and extractions in one setting, what explaining that the sum of restorations and extractions cannot be 100%.

Why not add this row into the table?

8.2 The sentence above 3.1: The majority of children with other diagnosis then dental caries were healthy 224 (97.82%).

Cannot be understood, grammar mistake.

Author Response

Dear reviewer. Thank you very much for all comments and suggestions. I did try to comply with them.

1.    The study was approved by the Ethical Committee, Faculty of Medicine, Charles University in Hradec Kralove.

I      tried to explain the aim of the study better. We though, based on our      clinical experience, that medically compromised children have received      more frequent and more extensive dental care under GA. We wanted to have      some objective data. This study comes up from country with different system of health care.      We do not have any public health service where patients with special needs      are regularly checked and medically compromised children may not achieve      dental treatment in time.

3.    It was designed as the retrospective study. I agree we do not have any follow up. But we work on it and collect the data from the re-call recently.

4.    I agree. All children were treated. We tried to find out extend of treatment – how many fillings and extractions have been performed under GA and if medical conditions do have impact on treatment.

5.    I am afraid that categorization will not help. We recorded all treated primary teeth and all treated permanent teeth. Some children with mixed dentition received the treatment of both primary and permanent teeth in one setting (25.33%). It is difficult categorize exactly that primary teeth are present only in children from 1 to 5 years, mixed from 6 to 12 and so. There might be individual differences. One 6 year old child can have still primary dentition and another 5 year old child mixed dentition.

6.    You are definitely right. Our statistician did not comment it more.

7.    I tried to use better formulation if acceptable.

Round 2

Reviewer 2 Report

highlight your changes in red

the English language needs extensive revision, 

abstract: 

add the findings of the study, tests that you used, pa values,,, add the average age of the patient as mean(SD)

2nd line, 'psychological disabilities' revise what do you mean?

add the type of study, x-sectional? retrospective?

line 6, 'from which 62.01% were diagnosed for any systemic disease' revise what do you mean? 

line 7, 'Dental treatment of primary teeth was more commonly performed in healthy children, 65.52% in comparison to medically compromised children 58.45%.' this is not good, revise, 

Introduction

very short, failed to develop the idea of why authors needed to perform this study

develop your null hypothesis at the end, then in result section discuss if you accepted it, rejected it?

M&M

very long paragraph, section it, add subheadings, explain more

'2.1. Statistics:' change to '2.1. Statistical analysis'

the first paragraph in the result section should be relocated to the M&M section, summarise the table 1 in the text

table 3 and 4, what test did you use? chi-square or exact test, add this information,  then you need to add the numbers as well show as n(%)

check the references, there are many typos, you don't need doi! 

reference no 4, should be 'Tahmassebi JF'

Author Response

Dear reviewer. Thank you very much for all comments and suggestions. I did try to comply with them.

You are right in almost all of them and I tried to explain or revise them in the text (text in red).

One comment was difficult to explain, so I gave it here: Tables 3 and 4 – chi-quadrat was used because the assumption of minimal expected frequency for this test was fulfilled.

If we add absolute frequency the readers may be confused, what is 100%. If results are in %, it means that all medically compromised/healthy children were calculated as 100%.

Round 3

Reviewer 2 Report

there are many typos and grammar issues, please revise, I have looked at abstract just to show you how many issues are there!

abstract., add the mean (SD) for the age of subjects, you don't need to say'Level of significance was α = 0.05.', remove

revise '. Dental treatment of primary teeth was more commonly performed in healthy children, 65.52% in comparison to medically compromised children 58.45%.' to '. Dental treatment of primary teeth was more commonly performed in healthy children (65.52% ),  compared to medically compromised children (58.45%.), add p-value if it was significant, use the same approach in the text as well

the following sentence does not make sense 'The number of all medically compromised/healthy children was calculated as 100%. ' revise

correct In permanent dentition medically compromised children required more extractions and fillings (38.03%, 57.04%) to healthy ones (14.94%, 17.24%).' to 'In permanent dentition, medically compromised children required more extractions and fillings (38.03%, 57.04%), compared to healthy ones (14.94%, 17.24%).'

what do you mean by 'Any systemic chronic disease was diagnosed in 142 (62.01%), 87 (37.99%) were uncooperative healthy children.' revise, does not make sense

revise the conclusion, very poorly written, what is the clinical massage here?

introduction and discussion, , very short, not focused, very deficient

table 1 should be summarised in the text, then remove the table

table 3, 4  add the numbers and the % , not just %, add the degree of freedom, chi-square value and p values

Author Response

Dear reviewer.

Thank you very much for your comments and suggestions. Based on your kind recommendations I have revised the manuscript and tried to upgrade it as much as I could. The changes are highlighted. A have also worked on improvements to the English.

Regarding statistics, I have discussed this issue with the statistician. She does not recommend to use numbers, just %. 

Dear reviewer, I would highly appreciate if you accept the manuscript.

Thank you very much.

R. Koberova

This manuscript is a resubmission of an earlier submission. The following is a list of the peer review reports and author responses from that submission.

Round 1

Reviewer 1 Report

In the manuscript entitled: “The need of dental treatment under general anaesthesia in medically compromised and healthy children” the authors identified and quantified variables of dental treatment under GA in medically compromised and healthy children.

In their in study, the data were collected from medical records of children aged 1 to 18 years received their dental treatment under GA. The data regarding patient age, sex, general health and treatment need were analysed.

The authors found that the treatment needs of primary teeth was higher in healthy children, 65.52% in comparison with medically compromised children 58.45%. On the contrary, medically compromised children required more extractions and fillings (38.03%, 57.04%) in permanent dentition compared to healthy ones (14.94%, 17.24%).

The authors concluded that the higher prevalence of dental caries in medically compromised children and therefore need of treatment should encourage health professionals to create the specific programs to prevent oral health problems in these children.

Major comments:

In general, the idea and innovation of this study, regards the analysis of dental treatment under general anaesthesia in medically compromised and healthy children is interesting, because the role of dental treatment in medical compromized patients is validated but further studies on this topic could be an innovative issue in this field could be open an innovative matter of debate in literature by adding new information. Moreover, there are few reports in the literature that studied this interesting topic with this kind of study design.

The study was well conducted by the authors; However, there are some concerns to revise that are described below.

The introduction section resumes the existing knowledge regarding the important factor linked with medical compromised patients.

However, as the importance of the topic, the reviewer strongly recommends, before a further re-evaluation of the manuscript, to update the literature through read, discuss and cites in the references with great attention all of those recent interesting articles, that helps the authors to better introduce and discuss the aim of the study in light of some methods of detect caries and some conditions linked with caries: 1) Isola G, Cicciù M, Fiorillo L, Matarese G. Association Between Odontoma and Impacted Teeth. J Craniofac Surg. 2017 May;28(3):755-758. 2) Isola G, Matarese G, Cordasco G, Rotondo F, Crupi A, Ramaglia L. Anticoagulant therapy in patients undergoing dental interventions: a critical review of the literature and current perspectives. Minerva Stomatologica 2015, 64:21-46. 3) Morsczeck, C.; Reichert, T.E. Dental stem cells in tooth regeneration and repair in the future. Expert Opin Biol Ther 2018, 18, 187-196.

The authors should be better specified, at the end of the introduction section, the rational of the study and the aim of the study with the null hypothesis. In the material and methods section, should better clarify how was performed the treatment of primary dentition. Moreover, specify if data were normalized or not. Please specify if was used a test unit.

The discussion section appears well organized with the relevant paper that support the conclusions, even if the authors should better discuss the relationship between anticoagulants and bleeding. The conclusion should reinforce in light of the discussions.

In conclusion, I am sure that the authors are fine clinicians who achieve very nice results with their adopted protocol. However, this study, in my view does not in its current form satisfy a very high scientific requirement for publication in this journal and requests a revision before publication.

Minor Comments:

Abstract:

-        Better formulate the introduction section by better describing the aim of the study

Introduction:

-        Please refer to major comments

Discussion

-        Please add a specific sentence that clarifies the results obtained in the first part of the discussion

-        Page 5 last paragraph: Please reorganize this paragraph that is not clear

Reviewer 2 Report

Revise the abstract, the objective of the study is not clear,

many grammar issues, do not start sentences with figures' 62.01% of children from current study,,,,'

what is the novel finding of this study? explain more, how did you come up with this conclusion 'The higher prevalence of dental caries in medically compromised children and therefore need of treatment should encourage health professionals to create the specific programs to prevent oral health problems in these children.'

very long M&M section, you need to add subheadings for variables you measures, study type, sample size calculation,,,

again very poorly written introduction and discussion

Reviewer 3 Report

Thank you for the invitation to review this manuscript. However, this manuscript is not well prepared and lots of terms and research concepts are wrong.

First and foremost, the study aim and design are unclear and may be incorrect.

1.     The study aim: It’s hard to understand the aim of the study: The purpose of this study was to identify and quantify variables of dental treatment under GA in medically compromised and healthy children. What does ‘variables’ refer, risk factors? Reasons? After reading the manuscript, it is still confused. Besides, it’s also hard to understand ‘quantify variables’. The conclusion didn’t match or answer the aim of this study.

2.     The study design: The age range in the study included infants, preschoolers, children and teenagers whose dental condition varied significantly. Moreover, the author mentioned that the study design was ‘retrospective’ (Pg2, first sentence of material and method) and ‘cohort’ (Pg 4, last sentence of the result). However, this study didn’t have any follow-up period and it cannot be a retrospective cohort study. Because of the misunderstanding of the design of the study, the discussion was incorrect (the first sentence of the discussion on Pg 7). Also, the following reporting guideline is also inappropriate (first sentence in the materials and methods).

There are also some important points need to be addressed.

Title: This title is unappropriated. If this study wants to describe the treatment need under GA, it should report the ‘need’ (the prevalence, the severity of the oral disease) between these two groups of children.

Abstract

1.     What is the statistic method adopted in this study?

2.     The data of the abstract is hard to understand because of the unclear measuring outcome.

For example, how to define treatment need? As all of the recruited children had received treatment under GA, what does treatment need refer in this manuscript?

Besides, the data fails to depict the whole picture of the situation. What kind of treatment under GA was included except filling and extraction (38%+57% is less than 100% and 15%+17% is also less than 100%)?

Background

1.     What is the research gap?

2.     The aim of this study is hard to understand.

Method:

1.     Have this study obtained the approval of IRB? If so, please add the information.

2.     Why choose 4 year (2015-2018)? And why choose this age group (Pg 2)?

3.     What does extensive oral health problem mean in the inclusion criteria? Need to give full name of ICD 10 and ASA.

4.     What are treatment need and general health? How to define them?

5.     What do the authors refer to about qualitative data? Qualitative data can not be measured by number. It’s incorrect to say ‘Counts and relative counts were used for description of qualitative data.

6.     Is this sample size representative? Was there a sample size calculation?

7.     Use of chi-squared test is incorrectly used.

Results

1.     Because the author described the treatment of primary teeth and permanent teeth, it is essential to categorize the data according to their age.

2.     ‘The most common dental condition diagnosed was dental caries, and its complications accounted for 81%. How about the rest?

3.     What is the distribution of the oral diseases (Caries? Trauma? Or others?) that needed treatment for GA for children in different group?

4.     ‘The treatment need of primary teeth, table 3, was higher in healthy children, 66% in comparison with medically compromised children 58%.’ (second sentence of 3.1). As the p-value of the comparison is not significant, this statement is not appropriate. 

5.     The data of Table 3 is confusing. It hard to understand what is being compared with (as the sum of percentage is not 100%, and the information of the comparison didn’t provide in the text). It appeared only restoration and extraction were the treatment items being studied. However the sum of percentage is not 100%. 

6.     The interpretation of the hypothesis testing is incorrect (For example: Health conditions have no impact on the need of primary teeth extractions (p= 0.289)).

7.     As the definition of ‘treatment need’ is unclear, the measurement and comparison of this outcome cannot be done.

Discussion

1.     Data should be reported in the result session instead of discussion session (Second paragraph of the discussion).

2.     The author failed to discussion about this study’s significance, meaning to this research field and the data reported. Little information was provided by this manuscript.

3.     How to control the bias?